# Gelatin–Curcumin Nanocomposites as a Coating for Implant Healing Abutment: In Vitro Stability Investigation

**Solmaz Maleki Dizaj** [1,†] **, Ali Torab** [2,†] **, Shadi Kouhkani** [2,†] **, Simin Sharifi** [1] **, Ramin Negahdari** [2,*] **, Sepideh Bohlouli** [3] **, Shirin Fattahi** [4] **and Sara Salatin** [1]

[1] Dental and Periodontal Research Center, Tabriz University of Medical Sciences, Tabriz 5166-15731, Iran
[2] Department of Prosthodontics, Faculty of Dentistry, Tabriz University of Medical Sciences, Tabriz 51368, Iran
[3] Department of Oral Medicine, Faculty of Dentistry, Tabriz University of Medical Sciences, Tabriz 51548-53431, Iran
[4] Department of Oral and Maxillofacial Pathology, Faculty of Dentistry, Tabriz University of Medical Sciences, Tabriz 5166-15731, Iran
* Correspondence: negahdari1358@gmail.com
† These authors contributed equally to this work.

**Abstract:** Regarding the importance of preventing peri-implantitis in dental implants, the current study aimed to coat a healing abutment with gelatin–curcumin nanocomposites, and the stability of this coating on the healing abutment was evaluated. A cell viability measuring test was used to determine the cytotoxicity of nanocomposites against dental pulp stem cells. To show the pattern of curcumin release from nanocomposites, drug dissolution apparatus two was applied. Then, 16 healing abutments were examined in vitro. Titanium healing abutments were coated with the gelatin–curcumin nanocomposite. The dip coating method was applied for coating and the consistency of coated cases was evaluated at intervals of one, 30, and 60 days after coating inside the simulated body fluid (SBF) solution. A scanning electron microscope (SEM) was used for investigating the microstructure and morphology of coatings, and an energy dispersive X-ray (EDX) was applied for determining the combination of the coating. Moreover, the healings were weighed before and after coating via an accurate digital scale with an accuracy of 0.0001. Finally, the data were analyzed using SPSS software. The prepared nanocomposite was non-cytotoxic against tested cells. The nanocomposite showed a relatively rapid release pattern in the first 10 days for curcumin. The release of curcumin from the nanoparticles continued slowly until the 30th day. The weight changes were statistically significant ($p$-value < 0.001) during this time. Based on the post hoc test, the weight between two times immediately after coating and 30 days after coating, and also one day after coating and 30 days after coating, was statistically insignificant. The results revealed that the coating of the gelatin–curcumin nanocomposite on the healing was successful and this consistency was kept for at least one month. It is necessary to investigate more evaluations in different fields of physicochemical, mechanical, and antimicrobial aspects for coated healing abutments.

**Keywords:** dental implant; healing abutment; gelatin; curcumin; tissue engineering; cell engineering; biomaterials

## 1. Introduction

In recent years, valid and predictable dental implants as a method for substituting lost teeth was confirmed with a primary success rate of approximately 94.6% [1]. Biological changes are unpreventable after losing teeth in hard and soft tissues. Increasing beauty expectations among patients has increased the popularity of dental implants to regenerate better function and beauty [2,3]. The healing abutment is used for creating an appropriate form of soft texture and an acceptable emergence profile to regenerate the soft textures better around the implant [4].

Preventing biological complications, such as peri-implant diseases, is important to achieve long-term success in implant-based treatment [5]. Peri-implant diseases are a collection of inflammatory reactions in the textures around dental implants and are known as the leading cause of long-term failure in implantology. Peri-implant diseases include two classifications: mucositis peri-implant and peri-implantitis. Mucositis peri-implant is an inflammatory lesion that is created around the implant mucus and can be converted into a peri-implant that involves the hard tissue if not treated [6]. In the more advanced stages, this disease can be observed in the form of redness, swelling soft tissue, gum recession, purulent exudate, and significant bone loss that results in implant laxity [7]. Thus, controlling plaque can help to prevent peri-implant diseases after surgery. However, regarding pain and swelling after surgery and trauma possibility as a result of brushing, patients cannot perform the techniques to control plaque. Further, considering that the healing abutment directly contacts the mouth environment and abutment implant, it probably increases the chances of creating peri-implantitis [5].

Plaque permanence after surgery increases pain, swelling, and trauma danger during brushing; thus, an antibacterial coating on the healing abutment can be beneficial to prevent these complications [8]. Investigations that have been performed for peri-implant treatment have concluded that disinfecting the infected surface of the implant with chemical factors is not sufficient (for example, saline, chlorhexidine, citric acid, and phosphoric acid with 35%) because they cannot completely eliminate biological survival, and mechanical debridement (such as airborne particle abrasions, laser, and titanium brush) can only remove part of biological survival [5,9]. Modifying titanium surfaces with an antibiotic coating, antimicrobial peptides, and a coating of chemically synthesized nanoparticles (such as Ag, Cu, and Zn) have been effective in various studies [5,10,11].

Some recent studies also stated that the existence of pathogens is essential but not sufficient for the development of peri-implant diseases. According to these reports, the osteo-immunoinflammatory mediators produced by the host response exert an essential impact on the breakdown of peri-implant tissue. The study of inflammatory mediators using sulcular fluid sampling can predict peri-implant diseases much earlier than through clinical appearance or radiographic changes [12,13].

A method for creating an antimicrobial coating on implant surfaces is nanomaterial coatings [14]. Nanomaterials have been widely applied in dentistry in recent years; for example, they can be applied in producing dental material, bonding systems, implant surfaces of veneer, mouthwash, and endodontic sealers [15]. Some nanoparticles, such as copper, zinc, magnesium, silver, and gold, have antimicrobial properties and can be used as protective covers on the implant surface and healing abutment. This antimicrobial property arises from the increased level and direct contact of nanoparticles with bacterial cell walls [16].

Gelatin is a natural biopolymer that is widely used in pharmacology and medication as a carrier of the delivery system of drugs and bandages for healing wounds because it has biological experience, biocompatibility, and does not stimulate the immune system [17].

Curcumin is a plant-based ingredient that has antimicrobial properties. Curcumin with the chemical formula [1,7-bis(4-hydroxy-3-methoxyphenyl)-1,6-heptadiene-3,5-dione] is a phenolic compound in the curcumin longa plant [18]. Curcumin has been widely used for orthodontics, periodontal diseases, mouth cancer medication, and other dentistry applications. In addition to the numerous therapeutic properties, the antimicrobial, antioxidant, wound healing, and anti-cancer properties have provided appropriate conditions for conducting this study. Thus, the improvement of dental implant properties using curcumin-based materials has encouraged researchers to focus on this substance [5]. Sahely et al. reported that titanium oxide nanotubes coated with curcumin have an antimicrobial effect against staphylococcus aureus and *Escherichia coli* in bone implants [15]. In addition, the conducted studies revealed no significant complication on implants coated with curcumin in terms of local and systemic toxicity [19].

Despite the good features of curcumin, its bioavailability is low and the main method for eliminating this case and increasing its bioavailability is using the nanoparticles of this substance [20]. The size of particle diffusion is one of the most important features of nanocomposites, as it influences the release, stability, and bioactivity of nanocomposites and also affects cell interactions [21]. Mandrol et al. investigated the influence of curcumin on human dental fibroblasts and reported that using nanocurcumin directly decreases cell toxicity to a greater extent than using curcumin [22].

Using gelatin as a curcumin carrier can improve the curcumin solution, resistance, and the control of its release [17,23]. Studies have reported that a gelatin–curcumin composite can increase surface interactions. Curcumin is slowly released when gelatin–curcumin particles are connected to the surface. Therefore, the unexpected toxicity caused by curcumin's high density is decreased [24]. Nago et al. concluded that gelatin–curcumin can overcome the bioavailability of curcumin, increase curcumin efficiency (due to the smaller size of nanocurcumin), and improve the pervading and releasing of the drug [25].

The current study aimed to investigate the stability of a healing abutment coating with gelatin–curcumin nanocomposites to determine its role in preventing peri-implant in dental implants, by using an antimicrobial coating with appropriate features to prevent this complication.

## 2. Materials and Methods

### 2.1. Nanocomposite Preparation

Gelatin–curcumin nanocomposite suspensions were obtained before the coating procedure. First, a gelatin (Sigma Aldrich, St. Louis, MO, USA) solution with a weight of 8% in acetic acid was obtained at room temperature. Approximately 5% of nanocurcumin powder (the average particle size of 95 nanometers purchased from Alborz Co., Tehran, Iran) was added to the gelatin solution. Suspensions were stirred on a magnetic stirrer for 24 h at room temperature. Glutardialdehyde (1%) was used for the gelatin cross-linking.

### 2.2. Coating Method

Titanium healing abutment with a 4.5 mm diameter and height of 4 mm was used from the brand Medimecca CEO- Geumcheon-gu, Seoul, Korea (CIS). They were cleaned for 20 min in acetone ultrasonic and then were kept in 70% ethanol solution for 20 min, and finally for 15 min in distilled water. Dip coating was used for coating which included 3 phases [14].

- Immersing healing abutments in nanocomposite suspensions.
- Setting the dip coating machine.
- Solvent evaporation (Figure 1).

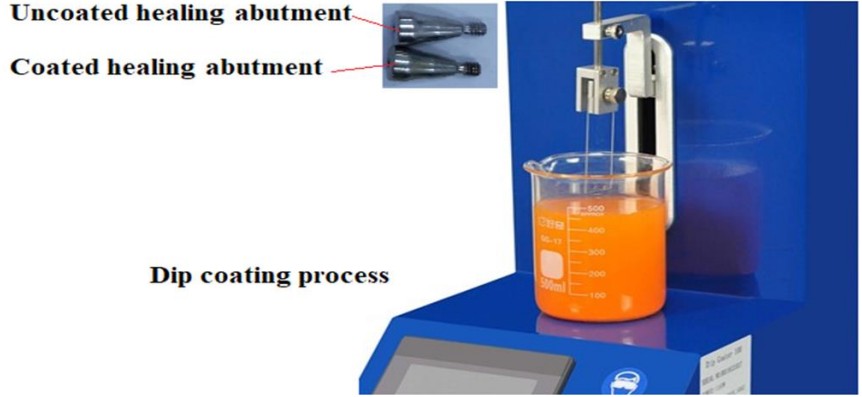

**Figure 1.** Dip coating process.

A scanning electron microscope (SEM) was used for investigating the substructures and coating morphology, and a transmission electron microscope transmission electron microscopy (TEM) for determining the presence of nanoparticles in the nanocomposite matrix. An energy dispersive X-ray (EDX) (Shimadzu, Kyoto, Japan) was used for determining the combination of the coating.

### 2.3. Assessment of the Cytotoxicity

A 3-(4,5-dimethylthiazol-2-yl)-2,5-diphenyltetrazoliumbromide (MTT)-based test was used to determine the cytotoxicity of nanocomposites (cell viability) according to ISO 10993-5 [26] against dental pulp stem cells. Dental pulp stem cells were bought from Shahid Beheshti University (Tehran, Iran). The disks of nanocomposites with a diameter of 5 mm were placed in the bottom of the wells and then the cells were cultured in a single layer in Dulbecco's Modified Eagle Medium (DMEM) containing serum and antibiotics. After 72 h, the cells were washed and incubated with 200 microliters of culture medium along with 50 microliters of MTT solution (2 mg/mL phosphate buffered saline (PBS)) for 4 h at 37 °C and away from light. After this period, the above solution was removed and 200 microliters of dimethyl sulfoxide (DMSO) and 25 microliters of Sorenson glycine buffer (containing glycine and sodium chloride) was added to each well. DMSO dissolves the colored crystals in the mitochondria of the cells, and after transferring the colored solution to the plate, their absorbance was read at 540 nm and the percentage of living cells was evaluated by comparing them with the control (cells grown without any material).

### 2.4. Curcumin Release Pattern from Nanocomposite

To show the pattern of curcumin release from nanocomposites, drug dissolution apparatus 2 was applied for 1 month. The drug dissolution apparatus 2 (or the rotating paddle device), is the most widely used method in drug dissolution testing, especially for nanomaterials. It includes 6 wells and a setting process including pH, temperature, and the paddle's rotation speed. Based on its standard, these parameters should be set according to the body's condition for a dissolution test of a drug (pH of 7.4, temperature of 37 °C, and stirring rate of 100 rpm) [27].

Phosphate buffer (300 mL) was poured into each well of the device (pH of 7.4, temperature of 37 °C, and stirring rate of 100 rpm). Five mg of the nanocomposite was poured into all wells. Liquid samples were taken from the wells every day (one milliliter) and the absorbance was read using a UV spectrophotometer. The sample taken from the wells was replaced with one milliliter of a new buffer medium. The amount of adsorption was converted to concentration. Then the cumulative release percent was plotted against time (day) for the release investigation.

### 2.5. Method of Stability Investigation

The following reagent-grade chemicals were used in preparing the simulated body fluid (SBF) solutions in deionized water [28]:

(1)    Sodium chloride (NaCl),
(2)    Potassium chloride (KCl),
(3)    Sodium hydrogen carbonate ($NaHCO_3$),
(4)    Magnesium chloride hexahydrate ($MgCl_2 \cdot 6H_2O$),
(5)    Sodium sulphate ($Na_2SO_4$),
(6)    Calcium chloride dihydrate ($CaCl_2 \cdot 2H_2O$),
(7)    Di-sodium hydrogen phosphate dihydrate ($Na_2HPO_4 \cdot 2H_2O$),
(8)    Tris (($CH_2OH)_3CNH_2$),
(9)    1 M HCl solution.

The coated samples were immersed and incubated in SBF solution for 1, 30, and 60 days. The SBF was exchanged every 2 days to have a better similarity with the liquids inside the body. The procedure was evaluated with an electron microscope after 1, 30, and

60 days. Further, the weight of coated and uncoated healing abutment was calculated after 1, 30, and 60 days.

### 2.6. Data Analysis

Mean and standard deviation (SD) were used for describing the data. The ANOVA test was run to compare the results using SPSS (version 16, IBM, New York, NY, USA) and the *p* value was considered $p < 0.05$.

### 2.7. Ethical Considerations

The current study was approved by the ethics committee, Tabriz University of Medical Sciences, with the code IR.TBZMED.VCR.REC.1400.547.

## 3. Results

### 3.1. Release Results

The nanocomposite showed a relatively rapid release pattern in the first 10 days for curcumin. The release of curcumin from the nanoparticles continued slowly until the 30th day (Figure 2a).

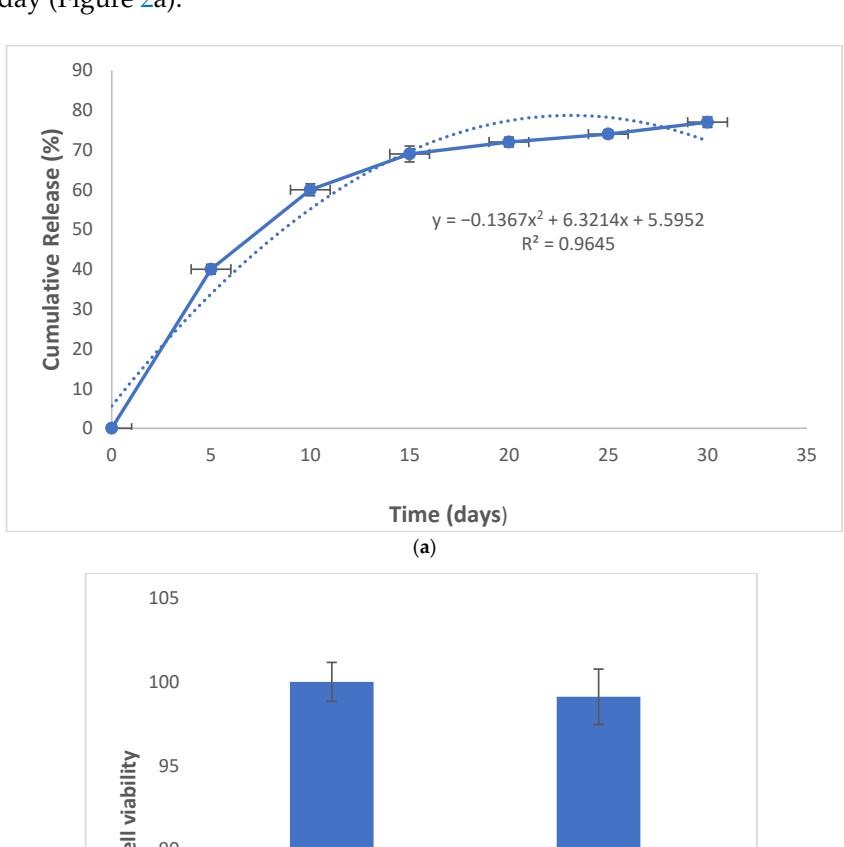

**Figure 2.** The release pattern of curcumin from nanocomposites (**a**) and the cytotoxicity of the composite against dental pulp stem cells (**b**).

### 3.2. Cytotoxicity

The prepared nanocomposite was non-cytotoxic against dental pulp stem cells (Figure 2b).

### 3.3. Stability Results

Figure 3 shows the electron microscope image (a), the EDX evaluation (b) of curcumin nanoparticles, the electron microscope image (c), and the EDX evaluation (d) of gelatin–curcumin nanocomposite.

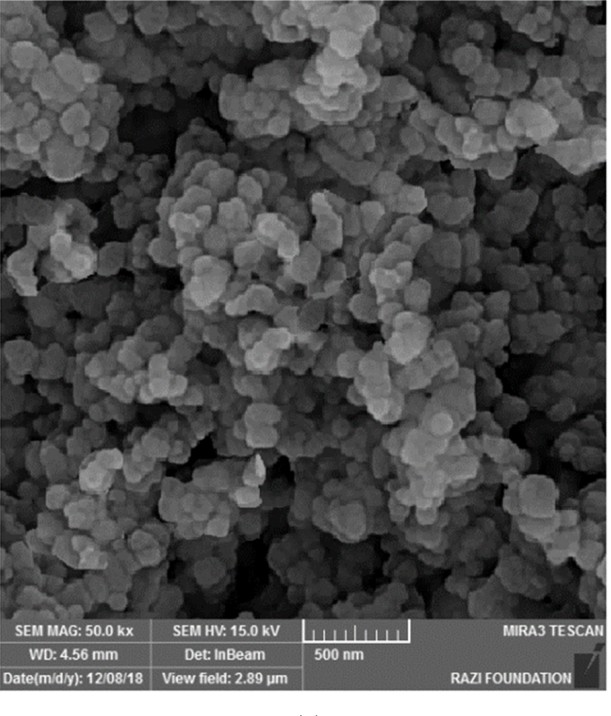

**(a)**

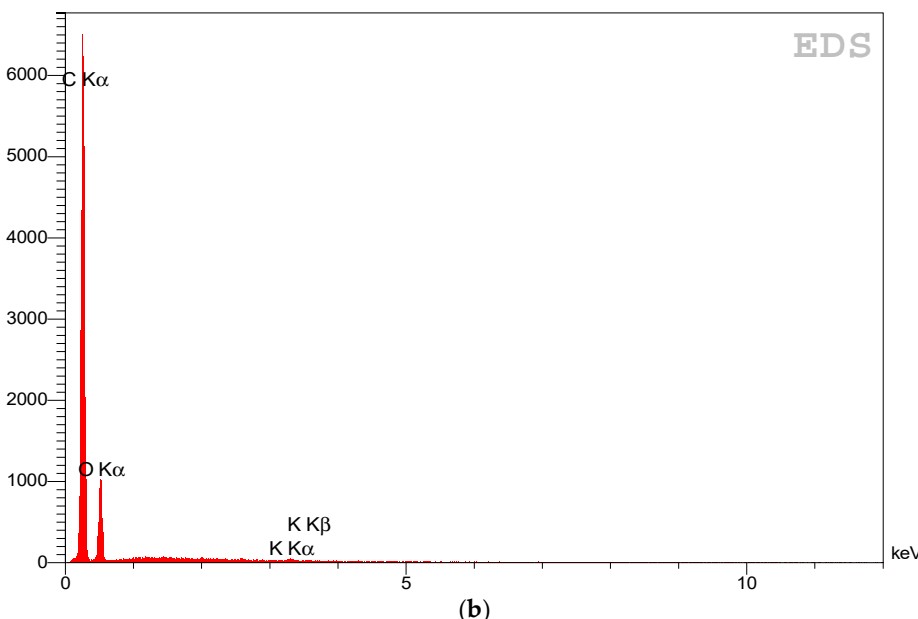

**(b)**

**Figure 3.** *Cont.*

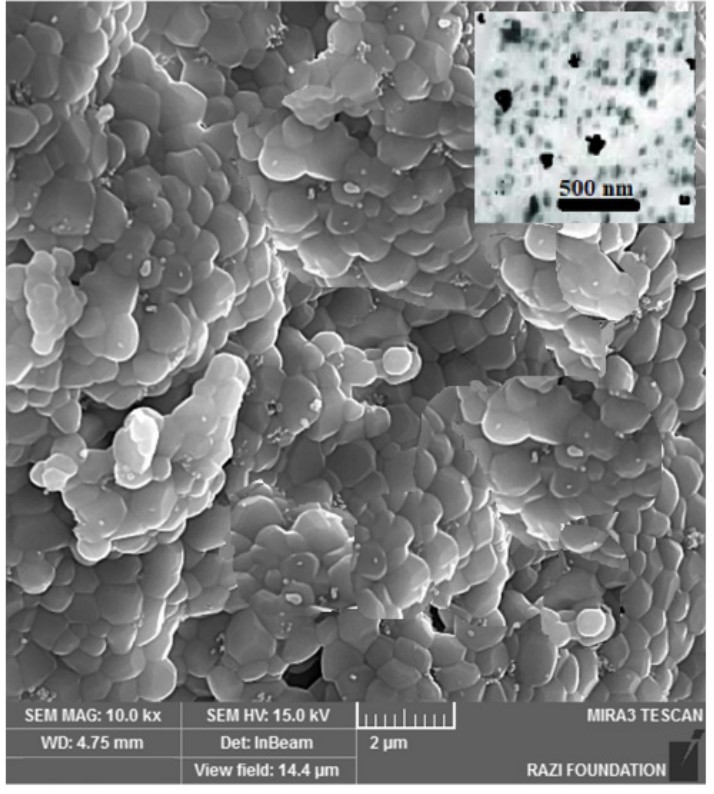

(**c**)

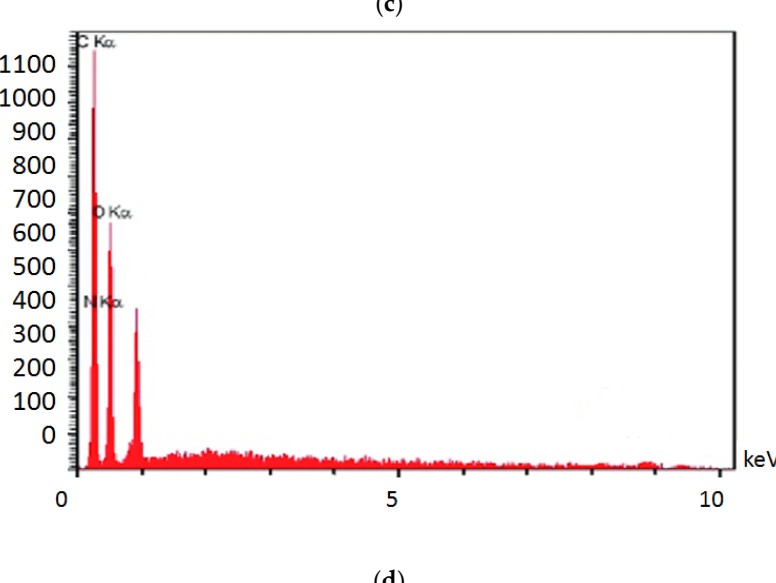

(**d**)

**Figure 3.** The electron microscope image (**a**), and EDX evaluation (**b**) of curcumin nanoparticles; the electron microscope image (SEM and TEM) (**c**), and EDX evaluation (**d**) of gelatin–curcumin nanocomposite.

Figure 4 shows the electron microscope image (a), the EDX evaluation (b) of the uncoated healing surface, the electron microscope image (c), the EDX evaluation (d) of the coated healing surface after one day, the electron microscope image (e), the EDX evaluation (f) of the coated healing surface after 30 days, the electron microscope image (g), and the EDX evaluation (h) of the coated healing surface after 60 days.

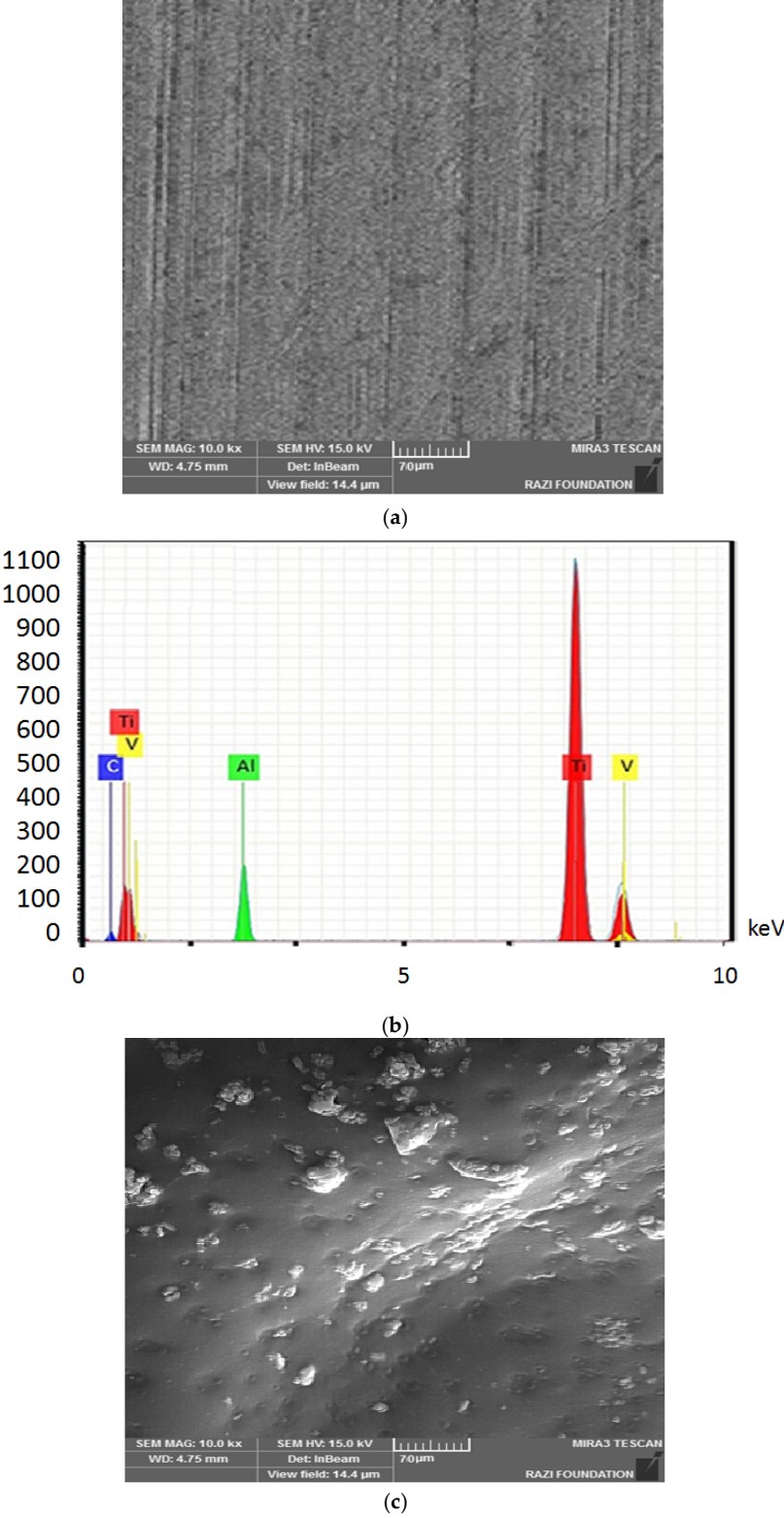

**Figure 4.** *Cont.*

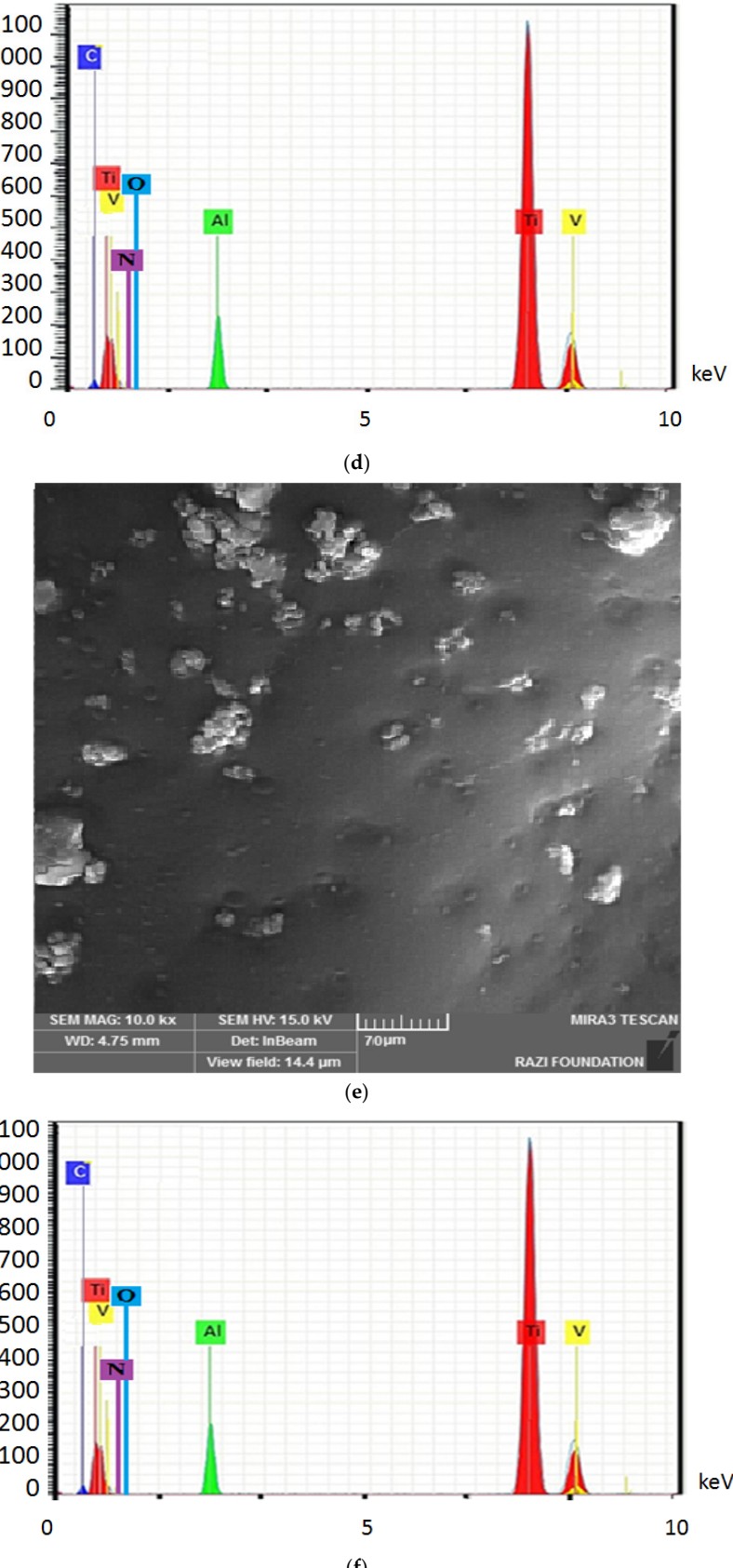

**Figure 4.** *Cont.*

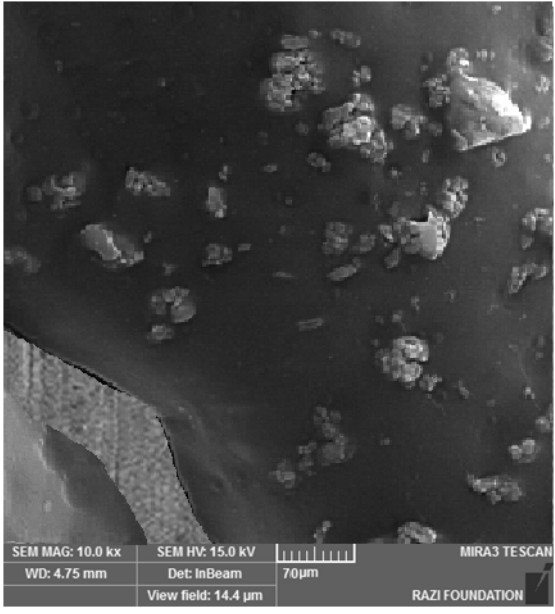

(**g**)

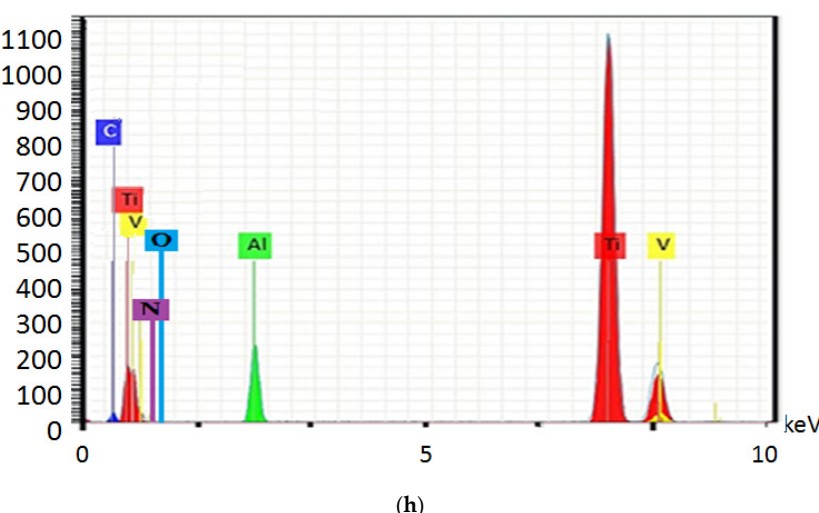

(**h**)

**Figure 4.** The electron microscope image (**a**), the EDX evaluation (**b**) of the uncoated healing surface, the electron microscope image (**c**), the EDX evaluation (**d**) of the coated healing surface after one day, the electron microscope image (**e**), the EDX evaluation (**f**) of the coated healing surface after 30 days, the electron microscope image (**g**), and the EDX evaluation (**h**) of the coated healing surface after 60 days.

An ANOVA test was run to compare weight change over time. The weight changes were statistically significant ($p$-value < 0.001) during time based on this analysis (Table 1).

**Table 1.** The weight of healings over time.

| Time | Number | Mean | SD |
|---|---|---|---|
| Primary weight | 16 | 0.3993 | 0.001 |
| Weight after coating | 16 | 0.4051 | 0.005 |
| One day after coating | 16 | 0.4050 | 0.005 |
| 30 days after coating | 16 | 0.4034 | 0.004 |
| 60 days after coating | 16 | 0.4013 | 0.003 |

Table 2 shows a pairwise comparison of the weight of samples between the studied times.

**Table 2.** The pairwise comparisons of the samples.

| Time | | Mean Difference | The Confidence Level of 95% for the Mean Differences | | *p*-Value |
|---|---|---|---|---|---|
| **(I)** | **(J)** | | **Low** | **High** | |
| Primary weight | Immediately after coating | −0.006 | −0.0087 | −0.0028 | 0.001 |
| | One day after coating | −0.006 | −0.0087 | −0.0027 | 0.001 |
| | 30 days after coating | −0.0040 | −0.0067 | −0.0015 | 0.001 |
| | 60 days after coating | −0.0020 | −0.0038 | −0.0002 | 0.021 |
| Immediately after coating | One day after coating | 0.0001 | 0.0000 | 0.0001 | 0.034 |
| | 30 days after coating | 0.0017 | −0.0003 | 0.0036 | 0.135 |
| | 60 days after coating | 0.0040 | 0.0015 | 0.0060 | 0.001 |
| One day after coating | 30 days after coating | 0.0016 | −0.0004 | 0.0036 | 0.167 |
| | 60 days after coating | 0.0040 | 0.0014 | 0.0059 | 0.001 |
| 30 days after coating | 60 days after coating | 0.0020 | 0.0009 | 0.0033 | 0.001 |

A Sidak test revealed that the weight between two times immediately after coating and 30 days after coating was not statistically significant. Moreover, the same happened for one day and 30 days after coating while the mean difference between both selected times was significant.

## 4. Discussion

The prepared nanocomposite was non-cytotoxic against tested cells. According to ISO 10993-5, which sets the rules for the biocompatibility of medical materials, the acceptable level of cytotoxicity for medical masks is as follows:

If the number of living cells is ≥80%, the sample is free of cytotoxicity.

If the number of live cells is >80%, the sample has moderate cytotoxicity.

If the number of living cells is >40%, the sample has high cytotoxicity [26].

The nanocomposite also showed a two-stage release pattern; a relatively rapid release pattern in the first 10 days for curcumin and a continuous release until the 30th day. The pattern of rapid release of curcumin in the outer surface of the composites and continuous release in the sixth to 35th days related to the nanoparticles are more internal and may have electrostatic interactions with the gelatin matrix and their release takes time [29]. Other investigators have also reported similar release patterns for curcumin-loaded dental materials [30,31]. Indeed, to achieve a slow-release behavior, either the sustained-release polymers are needed or a suitable cross-linker is used [32]. In a study by Saadipour et al. curcumin containing polycaprolactone and gelatin-blending fibers had a similar sustained-release pattern within 35 days. Using poly-caprolactone as a sustained-release polymer extended the release of curcumin [33]. In a study by Rubini et al., curcumin-containing gelatin films exhibit sustained curcumin release due to the existence of glutardialdehyde [34]. In our study, glutardialdehyde (1%) was used for gelatin cross-linking. This is the reason for the slow-release profile of curcumin from the nanocomposite after day 10. In a study by Sharifi et al., curcumin was released from the hydroxyapatite–gelatin nanocomposite until 14 days. According to the authors, the initial fast release of curcumin can be due to the diffusion process and the sustained release is due to the degradation of the hydroxyapatite–gelatin matrix [35]. They did not use a sustained release polymer or cross-linker. In this case, the slow-release pattern was observed just for 14 days. The existence of inorganic hydroxyapatite blending with gelatin can be the reason for the slow release.

Based on the results from the SEM, curcumin nanoparticles had a spherical morphology with particle sizes below 100 nanometers. The images of the gelatin–curcumin nanocomposite signified that curcumin nanoparticles have been diffused in the composite gelatin body (TEM results). Regarding uncoated healing, the images indicated a relatively smooth surface with natural unevenness. Regarding coated healing (one day, 30 days, and 60 days), the images showed nanocomposite coating success on the healing body. The images reported that curcumin nanoparticles were dispersed relatively homogeneously on the gelatin matrix. A gathering of nanoparticles on the gelatin surface at different points is observed. There was no significant difference between the first day and the 30th day; however, gathering nanoparticles were decreased in morphology by approximately the 60th day and a section of the gelatin matrix had been removed. Labias et al. reported that a carbon nanostructure coating has a different dispersion on dental implants in the different sections of the implant [36]. Of course, the difference between our research and this group is that coated healing has been carried out in the current study.

The EDX results showed that all the elements of curcumin, gelatin, and healing abutment were present. According to the EDX results, the element percentage of curcumin and gelatin had decreased on the 30th and 60th days rather than the first day. This indicates that curcumin and gelatin are released from the coated healing to the SBF environment after 30 and 60 days. The element analysis result of Labias et al. showed that element distribution percentages are different in various sections of the implant. The reasons for the difference can be differences in the used substances, differences in the coating method, and differences in the used implant [36].

An accurate digital scale with an accuracy of 0.0001 was used in this study to weigh the coated healings. The significant difference in uncoated healing in every four groups was observed immediately, one day, 30 days, and 60 days after coating. The significant increase in healing weight after coating showed the successful coating of healing with the gelatin–curcumin nanocomposite. The statistical result of the post hoc test indicated that there is a significant difference in terms of weight between all the groups (immediately, one day, and 30 days) with the 60 day group after veneering. There was no significant difference between the other groups. This finding shows that the nanocomposite coating of the healings was stable for at least 30 days. Fajer et al. investigated the permanence of an implant coating of titanium alloys coated with collagen at times of two, four, and six weeks, and concluded that the weight reduction for the coated implant was significant between two and six weeks. This means that the used coating had stability for at least four weeks [37].

## 5. The Limitations

This study was an in vitro investigation. Any probable toxicity of the new coating should be tested in animal models in future studies before any clinical tests. The antimicrobial and antibiofilm mechanisms for the prepared nanocomposite should be studied to verify its exact function.

## 6. Conclusions

The study findings revealed that the gelatin–curcumin nanocomposite was non-cytotoxic against dental pulp stem cells and showed a two-stage continuous release pattern for curcumin until the 30th day. The nanocomposite coating of healings lasted for at least one month. The prepared coating needs requires greater evaluation in different fields concerning its physicochemical, mechanical, and antimicrobial aspects, which we will address in future studies.

**Author Contributions:** Conceptualization, S.M.D. and R.N.; data curation, R.N. and S.S. (Simin Sharifi); formal analysis, S.M.D., S.K., R.N. and S.F.; funding acquisition, S.M.D. and S.S. (Sara Salatin); investigation, A.T.; methodology, S.K. and R.N.; project administration, A.T., S.B., S.S. (Simin Sharifi) and S.S. (Sara Salatin); resources, S.B.; supervision, R.N.; validation, A.T.; visualization, S.M.D.; writing—original draft, S.M.D. and R.N.; writing—review and editing, A.T., S.K., S.B., S.S. (Simin Sharifi), S.F. and S.S. (Sara Salatin). All authors have read and agreed to the published version of the manuscript.

**Funding:** This study was based on a thesis (Grant number 68412) registered at the Faculty of Dentistry, Tabriz University of Medical Sciences, Tabriz, Iran. It was financially supported by the Vice-Chancellor for Research at Tabriz University of Medical Sciences, Tabriz, Iran that is greatly acknowledged.

**Institutional Review Board Statement:** This study was approved by the Ethics Committee of the Tabriz University of Medical Sciences (Ethical code: IR.TBZMED.VCR.REC.1400.547).

**Informed Consent Statement:** Not applicable.

**Data Availability Statement:** The raw/processed data required to reproduce these findings can be shared after publication by request from the corresponding author.

**Conflicts of Interest:** The authors declare no conflict of interest.

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
