# Peer review of "Gelatin–Curcumin Nanocomposites as a Coating for Implant Healing Abutment: In Vitro Stability Investigation"

_clinpract, doi:10.3390/clinpract13010009_

Round 1

Reviewer 1 Report

Dear Authors, 

you made a great work. 

However, some improvements are mandatory before acceptance. 

The paper is a in vitro study on the Gelatin-curcumin nanocomposites as a coating for implant healing-abutment.

The Authors made a great work in terms of methodology and the paper sounds scientific and well written.

However, some improvements are mandatory before acceptance.

The abstract is well written, complete and summary in its various aspects. The keywords are complete and appropriate. Please check Keyword “bi-omaterials”.

Please check the text for double spaces, "-", and punctuation.

In the introduction, please check: • “Thus, applying methods of controlling plaque can help prevent creating peri-implement after surgery.”

• In particular, I suggest to the authors to consider in the introduction the current possibility offered by the analysis of inflammatory mediators to predict peri-implant damage with atraumatic examinations of sulcular fluid sampling and PICF, capable of orienting towards a pathological condition much earlier with respect to the appearance of clinical or radiographic changes, as indicated by: “Guarnieri R, Reda R, Zanza A, Miccoli G, Nardo DD, Testarelli L. Can Peri-Implant Marginal Bone Loss Progression and a-MMP-8 Be Considered Indicators of the Subsequent Onset of Peri-Implantitis? A 5-Year Study. Diagnostics (Basel). 2022 Oct 26;12(11):2599. doi: 10.3390/diagnostics12112599.”

Materials and methods are clear and well explained. Different aspects are analyzed with a dedicated statistical test. The authors did a great job in the explication of all the variables identified and included in the study.

Results are easy to understand and comprehensive. All the studied characteristics were reported in tables which are clear and concise.

Discussion: this section is complete and evaluates the outcome of different papers present in literature. The overall is comprehensive, concise and complete in its various aspects.

Please check: “The authers reported that the initial fast release of curcumin can be due to the diffusion process and t”

Conclusions are concise and clear.

Bibliography is formatted respecting the journal’s requirements and no improper citations are evidenced.

Figures and labels are clear and easy to comprehend.

Author Response

Dear Authors, 

you made a great work. However, some improvements are mandatory before acceptance. The paper is a in vitro study on the Gelatin-curcumin nanocomposites as a coating for implant healing-abutment. The Authors made a great work in terms of methodology and the paper sounds scientific and well written. However, some improvements are mandatory before acceptance.

Thanks a lot for your valuable comments.

The abstract is well written, complete and summary in its various aspects. The keywords are complete and appropriate. Please check Keyword “bi-omaterials”.

We checked it and corrected. Thanks.

Please check the text for double spaces, "-", and punctuation.

We checked it and corrected. Thanks.

In the introduction, please check: • “Thus, applying methods of controlling plaque can help prevent creating peri-implement after surgery.”

We checked it and corrected. Thanks.

• In particular, I suggest to the authors to consider in the introduction the current possibility offered by the analysis of inflammatory mediators to predict peri-implant damage with atraumatic examinations of sulcular fluid sampling and PICF, capable of orienting towards a pathological condition much earlier with respect to the appearance of clinical or radiographic changes, as indicated by: “Guarnieri R, Reda R, Zanza A, Miccoli G, Nardo DD, Testarelli L. Can Peri-Implant Marginal Bone Loss Progression and a-MMP-8 Be Considered Indicators of the Subsequent Onset of Peri-Implantitis? A 5-Year Study. Diagnostics (Basel). 2022 Oct 26;12(11):2599. doi: 10.3390/diagnostics12112599.”

It has been done. Thanks

Materials and methods are clear and well explained. Different aspects are analyzed with a dedicated statistical test. The authors did a great job in the explication of all the variables identified and included in the study.

Thanks

Results are easy to understand and comprehensive. All the studied characteristics were reported in tables which are clear and concise.

Thanks

Discussion: this section is complete and evaluates the outcome of different papers present in literature. The overall is comprehensive, concise and complete in its various aspects.

Thanks

Please check: “The authers reported that the initial fast release of curcumin can be due to the diffusion process and t”

We checked it and corrected. Thanks.

Conclusions are concise and clear.

Thanks

Bibliography is formatted respecting the journal’s requirements and no improper citations are evidenced.

Thanks

Figures and labels are clear and easy to comprehend.

Thanks

Reviewer 2 Report

I think the reasearch is very interesting, please just improve some aspects. 

The paper is a in vitro study on the Gelatin-curcumin nanocomposites as a coating for implant healing-abutment. Please check English style and punctuation in the whole text.

In the introduction:

• Please check the meaning of “. Thus, the symptoms of improv-ing dental implant 89

• properties in terms of using curcumin have encouraged researchers to focus on that in the future [5].

• I believe that to enrich the introduction from this point of view, the authors should consider the recent aspects of the study of inflammation of the peri-implant tissues, an element that rapidly indicates the progression of the peri-implant pathology before evident clinical or radiographic alterations. “Guarnieri R, Zanza A, D'Angelo M, Di Nardo D, Del Giudice A, Mazzoni A, Reda R, Testarelli L. Correlation between Peri-Implant Marginal Bone Loss Progression and PeriImplant Sulcular Fluid Levels of Metalloproteinase-8. J Pers Med. 2022 Jan 6;12(1):58. doi: 10.3390/jpm12010058.”

Materials and Methods section is completed and well described.

The results are clear, and if well organized they are easily readable and well summarized in complete tables and with particularly didactic figures.

In the Discussion: • Please describe the limitations of this study.

• Please check typos.

Conclusions are clear.

English is clear but should be improved.

Author Response

I think the reasearch is very interesting, please just improve some aspects. 

The paper is a in vitro study on the Gelatin-curcumin nanocomposites as a coating for implant healing-abutment. Please check English style and punctuation in the whole text.

Thanks for your valuable comments. We improved the manuscript. Thanks.

In the introduction:

• Please check the meaning of “. Thus, the symptoms of improv-ing dental implant properties in terms of using curcumin have encouraged researchers to focus on that in the future [5].

We checked and improved it. Thanks.

• I believe that to enrich the introduction from this point of view, the authors should consider the recent aspects of the study of inflammation of the peri-implant tissues, an element that rapidly indicates the progression of the peri-implant pathology before evident clinical or radiographic alterations. “Guarnieri R, Zanza A, D'Angelo M, Di Nardo D, Del Giudice A, Mazzoni A, Reda R, Testarelli L. Correlation between Peri-Implant Marginal Bone Loss Progression and PeriImplant Sulcular Fluid Levels of Metalloproteinase-8. J Pers Med. 2022 Jan 6;12(1):58. doi: 10.3390/jpm12010058.”

We applied your suggestion. Thanks.

Materials and Methods section is completed and well described.

Thanks

The results are clear, and if well organized they are easily readable and well summarized in complete tables and with particularly didactic figures.

Thanks

In the Discussion: • Please describe the limitations of this study.

It has been done. Thanks.

• Please check typos.

It has been done. Thanks.

Conclusions are clear.

Thanks

English is clear but should be improved.

We improved. Thanks

Reviewer 3 Report

The purpose of this research was to test the stability of a gelatin-curcumin nanocomposites covering on a healing abutment. The cytotoxicity of nanocomposite towards dental pulp stem cells was evaluated using a cell viability measurement technique. Drug dissolution apparatus 2 was used to demonstrate the release profile of curcumin from the nanocomposite. Sixteen different types of healing abutment were then subjected to in-vitro testing. The gelatin-curcumin nanocomposite was applied to titanium healing abutments. Coating was accomplished by dipping the cases in the SBF solution, and the consistency of the coating was assessed at 1, 30, and 60 days. As a result, the coating's microstructure and morphology were studied using a scanning electron microscope (SEM), and the coating's composition was ascertained using an energy dispersive X-ray (EDX). Moreover, a digital scale precise to within 0.0001 was used to weigh the healings before and after coating. At last, SPSS was used to examine the information gathered. No cytotoxicity was observed with the produced nanocomposite. In the first 10 days, the nano-composite exhibited a somewhat rapid release pattern for curcumin. Curcumin continued to be steadily released from the nanoparticles until the 30th day. When comparing the weight before and after coating, as well as the weight one day after coating, there was no statistically significant difference, as determined by a post hoc test. The gelatin-curcumin nanocomposite covering was effective in promoting healing, with the effects lasting for at least a month. More assessments of coated healing-abutments in physicochemical, mechanical, and antibacterial realms are required.

1- There are many typo errors and language is confusing and various sentences have no fluency so complete editing is required. 

2-"Therefore, an scaning electron microscope (SEM) was used for investigating the microstructur"  in abstract "an" or "a"

3-More strong and recent references are required in introduction part to support the effects of curcumin.

4-2.2. Coating method: has no reference, is its done first time in history?

5: Result section is poorly described without any sub division of various studies so it should be written again in different subsections and all results should be corelated with previous studies from literature.

Author Response

1- There are many typo errors and language is confusing and various sentences have no fluency so complete editing is required. 

Many thanks for your valuable comments. We improved the manuscript.

2-"Therefore, an scaning electron microscope (SEM) was used for investigating the microstructur"  in abstract "an" or "a"

It has been corrected.

3-More strong and recent references are required in introduction part to support the effects of curcumin.

It has been done.

4-2.2. Coating method: has no reference, is its done first time in history?

We added the reference.

5: Result section is poorly described without any sub division of various studies so it should be written again in different subsections and all results should be corelated with previous studies from literature.

It has been improved.

Round 2

Reviewer 2 Report

Dear Authors, 

thank you for your precious work! I think the manuscript is now suitable for publication. 

Congratulations!

Reviewer 3 Report

Corrections are done as advised.